
# Areas and entropies in BFSS/gravity duality

**Tarek Anous,[1,2⋆], Joanna L. Karczmarek,[2†], Eric Mintun,[2‡],**
**Mark Van Raamsdonk[2§] and Benson Way[2,3¶]**

**1** Δ-Institute for Theoretical Physics & Institute for Theoretical Physics, University of
Amsterdam, Science Park 904, Postbus 94485, 1090 GL, Amsterdam, The Netherlands
**2** Department of Physics and Astronomy, University of British Columbia,
6224 Agricultural Road, Vancouver, B.C., V6T 1Z1, Canada
**3** Departamente de Física Quántica i Astrofísica, Institut de Ciènsies del Cosmos,
Universitat de Barcelona, Martí i Franquès, 1, E-08028 Barcelona, Spain

⋆ t.m.anous@uva.nl
† joanna@phas.ubc.ca
‡ eric.mintun@gmail.com
§ mav@phas.ubc.ca
¶ benson@icc.ub.edu

## Abstract

The BFSS matrix model provides an example of gauge-theory / gravity duality where the
gauge theory is a model of ordinary quantum mechanics with no spatial subsystems. If
there exists a general connection between areas and entropies in this model similar to the
Ryu-Takayanagi formula, the entropies must be more general than the usual subsystem
entanglement entropies. In this note, we first investigate the extremal surfaces in the
geometries dual to the BFSS model at zero and finite temperature. We describe a method
to associate regulated areas to these surfaces and calculate the areas explicitly for a
family of surfaces preserving $SO(8)$ symmetry, both at zero and finite temperature. We
then discuss possible entropic quantities in the matrix model that could be dual to these
regulated areas.


# 1 Introduction

There is increasing evidence that spacetime geometry is related in a fundamental way to the entanglement structure of the underlying degrees of freedom in quantum theories of gravity [1–5]. In the context of the AdS/CFT correspondence, this is manifested most clearly in the Ryu-Takayanagi formula [2,6] that relates the areas of extremal surfaces in the gravity picture to the entanglement entropy of spatial subsystems in the dual CFT. So far, this connection applies only to minimal-area extremal surfaces homologous to some boundary region.[1] It is interesting to ask whether other extremal surfaces (or more general classes of surfaces) have an interpretation in terms of entropy. Various investigations along these lines have appeared in the past, for example [9–12].

If the connection between geometry and entanglement and/or the area/entropy connection is truly fundamental, it should be expected to apply for any theory of quantum gravity, even one where the fundamental description has no spatial subsystems.

In this note, we begin an investigation of the possible connection between extremal surface areas and entropies in the BFSS matrix model [13], which provides an example of gauge theory / gravity duality for which the gauge theory is simply a quantum mechanical theory with no spatial subsystems or natural decomposition of its Hilbert space into tensor factors. In the 't Hooft limit, this is dual to a ten-dimensional gravitational theory (type IIA string theory) on geometries which are asymptotic to the near-horizon D0-brane geometry [14]. The gravitational picture is slightly more complicated compared with typical examples of AdS/CFT since the geometry becomes strongly curved in the asymptotic region, so the supergravity description breaks down. However, we will focus on gravitational observables that are localized to the region in which classical gravity provides a good description.

We consider both the vacuum state of the model and finite temperature states, dual to ten-dimensional D0-brane black holes. The horizons of these black holes are extremal surfaces, and their area is expected to correspond to the entropy of the corresponding thermal state in

---

[1]More recently, it has been suggested that the areas of partial extremal surfaces may be related to a quantity called the entropy of purification [7,8].

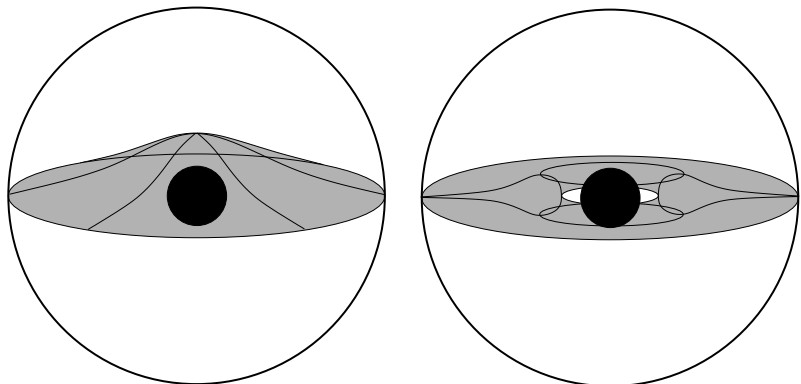

Figure 1: Extremal surfaces in near-horizon black D0-brane geometries. Left: surfaces with ball topology. Right: surfaces with cylinder topology.

the BFSS model. In fact, this has been checked via direct numerical simulation of the matrix model [15–21]. Our goal in this work will be to understand the other extremal surfaces in these geometries, to define finite, regulated areas that we can associate to these surfaces, and to discuss possible entropic quantities in the underlying model that could correspond to these areas. If the correct entropic quantities can be identified, it may be possible to calculate them directly via numerical simulation of the matrix model, providing a new detailed test of the AdS/CFT correspondence and the entropy/area connection.

The outline of the paper is as follows. In section 2, we review the definition of the matrix model and the dual gravitational description of its vacuum and thermal states. These dual geometries preserve $SO(9)$ rotational invariance. In section 3, we study extremal surfaces preserving $SO(8)$ invariance. These come in two varieties, as shown in figure 1: first, we can have surfaces with the topology of a ball that are bounded by an equator of the $S^8$ at infinity and have a single point of minimum radial coordinate. More generically, we can have surfaces with the topology of a (finite) cylinder that start at some equator of the $S^8$, reach their minimum radial coordinate on an $S^7$ and then go back out to infinity, asymptoting to the same equator on which they started.

In section 4, we focus on the surfaces with ball topology and define a regulated area for these surfaces. We numerically compute this area as a function of mimimum radial coordinate for surfaces in the vacuum geometry and the black hole geometries dual to thermal states.

In section 5, we consider more general extremal surfaces that break the $SO(8)$ symmetry. We would like to understand how these are parameterized so that we can look for entropic quantities in the matrix model that are parameterized in the same way.

In section 6, we discuss possibilities for entropic quantities in the BFSS matrix model that might correspond to the regulated areas computed in section 3. We review the notion of an entropy associated to a subalgebra and also consider the possibilities of entropies that could be associated to other subsets of observables.

The question we consider in this paper is related to the question of whether there is a CFT interpretation for the areas of extremal surfaces in AdS$^5 \times S^5$ that are wrapped on AdS$^5$ and are codimension one on $S^5$. This has been discussed previously in [22–24]. The D0 brane metric which we study in this paper is conformal to AdS$_2 \times S^8$, and one could consider going to the dual frame of [25] to make closer connection to the AdS$^5 \times S^5$ case, however this will not be our focus.

The recent work [26] that appeared while this paper was in preparation also considers the emergence of geometry and the connection between areas and entropies in matrix models (see also [27–31] for related considerations). For a discussion on entanglement and its potential

dual in the $c = 1$ model, see [32, 33].

# 2 Background

## 2.1 The BFSS model

The BFSS matrix model is a quantum mechanical system defined by the Hamiltonian

$$H = \text{tr}\left( \frac{1}{2} P^i P^i - \frac{g_{YM}^2}{4} [X^i, X^j]^2 + g_{YM} \Psi^\alpha \gamma_{\alpha\beta}^i [X^i, \Psi_\beta] \right), \tag{1}$$

together with the constraint that the states should be invariant under the $U(N)$ symmetry of the model. Here, $X^i$ and $P^i$ are a set of nine $N \times N$ Hermitian matrices of bosonic operators with commutation relations

$$\left[ X_{ab}^i, P_{cd}^j \right] = i \delta^{ij} \delta_{ad} \delta_{bc}, \tag{2}$$

with $i, j = 1 \ldots 9$. $\Psi_\alpha$ is a Hermitian matrix built from 16 component spinors with anticommutation relations

$$\left\{ (\Psi_\alpha)_{ab}, (\Psi_\beta)_{cd} \right\} = \delta_{\alpha\beta} \delta_{ad} \delta_{bc}, \tag{3}$$

with $\alpha, \beta = 1 \ldots 16$. The matrices $\gamma^i$ can be taken to be real and symmetric and satisfy the $SO(9)$ Clifford algebra: $\{\gamma^i, \gamma^j\} = 2\delta^{ij}$.

The parameter $g_{YM}$ has dimensions of $M^{\frac{3}{2}}$, so the 't Hooft coupling $g_{YM}^2 N$ has dimensions of $M^3$. In the 't Hooft limit, we consider $N \to \infty$ and focus on the physics at energies $E \sim (g_{YM}^2 N)^{\frac{1}{3}}$.[2] When considering thermal states, we take energy small enough so that $\lambda_{\text{eff}} \equiv g_{YM}^2 N / E^3 \gg 1$. This will ensure that there is a region in the geometry outside the black hole horizon for which supergravity provides a good description.

## 2.2 Gravity dual

The BFSS model at temperature $T$ corresponds in the gravity picture to the near-horizon D0-brane solution at finite temperature. Defining $\ell_s = \sqrt{\alpha'}$, the string frame metric for this solution takes the form

$$\frac{ds^2}{\ell_s^2} = -\frac{r^{\frac{7}{2}}}{\sqrt{\lambda d_0}} f_0(r) dt^2 + \frac{\sqrt{\lambda d_0}}{r^{\frac{7}{2}}} \left( \frac{dr^2}{f_0(r)} + r^2 d\Omega_8^2 \right), \tag{4}$$

with dilaton

$$e^\phi = \frac{(2\pi)^2}{d_0} \frac{1}{N} \left( \frac{\lambda d_0}{r^3} \right)^{\frac{7}{4}} \tag{5}$$

and RR one-form gauge field potential

$$A_0 = \frac{N}{2\pi^2} \frac{r^7}{\lambda^2 d_0}, \tag{6}$$

where

$$f_0(r) = 1 - \frac{r_H^7}{r^7}, \qquad \frac{1}{T} = \frac{4}{7} \pi \sqrt{\lambda d_0} r_H^{-\frac{5}{2}}, \tag{7}$$

and

$$d_0 = 240 \pi^5. \tag{8}$$

---

[2] Note that this is different from the limit considered originally by BFSS to define the flat-space limit of $M$-theory; that limit focuses on energies that are of order $g_{YM}^{\frac{2}{3}} / N$.

The dimensionful parameter $\lambda$ can be identified with the 't Hooft coupling in the gauge theory, or related to string theory parameters as $\lambda = g_{YM}^2 N = g_s N/(4\pi^2 \ell_s^3)$. Also notice that the coordinate $r$ has units of $[\text{length}]^{-1}$.

We'd like to find the extremal surfaces in this geometry preserving an $SO(8)$. However, to calculate the entropy and evaluate extremal surfaces, we should be working in the Einstein frame metric, obtained by the replacement $g_{\mu\nu} \to g_{\mu\nu} e^{-\phi/2}$. This gives a spatial metric

$$ds_E^2 = C r^{-\frac{7}{8}} \left( \frac{dr^2}{f_0(r)} + r^2 d\Omega_8^2 \right), \tag{9}$$

where

$$C = \frac{\ell_s^2 N^{\frac{1}{2}} d_0^{\frac{1}{8}}}{2\pi \lambda^{\frac{3}{8}}}. \tag{10}$$

It is convenient to change variables to $r = (R^2/C)^{\frac{8}{9}}$. Then we have

$$ds_E^2 = \frac{256}{81} \frac{dR^2}{1 - \left( \frac{R_H}{R} \right)^{\frac{112}{9}}} + R^2 d\Omega_8^2, \tag{11}$$

where

$$R_H = \sqrt{C} r_H^{\frac{9}{16}} = 2^{\frac{13}{20}} 15^{\frac{7}{40}} 7^{-\frac{9}{40}} \pi^{\frac{3}{5}} \left( \frac{T}{\lambda^{\frac{1}{3}}} \right)^{\frac{9}{40}} N^{\frac{1}{4}} \ell_s. \tag{12}$$

In particular, at zero temperature (or for large $R$) this becomes the metric of a cone

$$ds_E^2 = \frac{256}{81} dR^2 + R^2 d\Omega_8^2. \tag{13}$$

**Validity of the supergravity approximation**

To understand where the type IIA supergravity approximation is valid, we need to look at the behavior of the string frame curvature (in string units) and the dilaton, requiring that both of these be small.

The string frame curvature is small when the radius of the $S^8$ in the geometry (4) is large in string units. This requires that

$$r \ll \lambda^{\frac{1}{3}}. \tag{14}$$

Requiring that the dilaton is small gives

$$r \gg N^{-\frac{4}{21}} \lambda^{\frac{1}{3}}. \tag{15}$$

Thus, for large $N$, type IIA supergravity provides a good approximation to the bulk physics when

$$\frac{1}{N^{\frac{4}{21}}} \ll \frac{r}{\lambda^{\frac{1}{3}}} \ll 1. \tag{16}$$

In terms of the $R$ coordinate, this gives

$$N^{\frac{1}{7}} \ll \frac{R}{\ell_s} \ll N^{\frac{1}{4}}. \tag{17}$$

Thus, in the large $N$ limit, we have a parametrically large range of the radial coordinate where type IIA supergravity provides a good approximation to the physics.

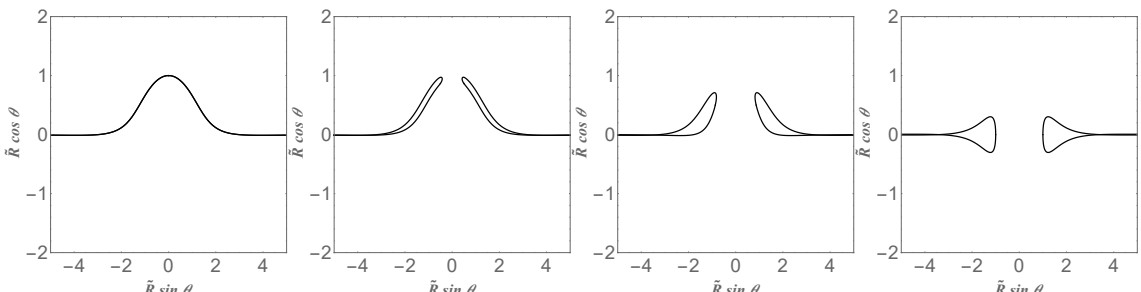

Figure 2: Extremal surfaces in $R, \theta$ plane with fixed minimum radius, with $\tilde{R} \equiv R \ell_s^{-1} N^{-\frac{1}{4}}$. The surfaces of revolution about the horizontal axis provide a picture of the complete extremal surfaces (where the extra $S^7$ is replaced by an $S^1$). We focus on surfaces of the first type, with disk topology. The surfaces with cylindrical topology are self intersecting.

**Entropy**

In terms of the $R$ coordinate, the entropy of the finite-temperature black hole states is given by

$$S = \frac{\omega_8 R_H^8}{4G_{10}}. \tag{18}$$

Using $G_{10} = 8\pi^6 g_s^2 \ell_s^8$ and $\omega_8 = 32\pi^4/105$, we find that in terms of the dimensionless temperature parameter $\hat{T} = T/\lambda^{\frac{1}{3}}$,

$$S = \left(\frac{15}{256}\right)^{\frac{2}{5}} \left(\frac{8\pi}{7}\right)^{\frac{14}{5}} N^2 \hat{T}^{\frac{9}{5}}. \tag{19}$$

## 3 Extremal surfaces

We would now like to investigate the extremal surfaces in these geometries. To begin, we find the codimension-one extremal surfaces in this spatial metric preserving an $SO(8)$ symmetry. Defining $\theta$ to be the angle on the $S^8$ from one of the poles fixed by this $SO(8)$, we can parameterize the surface by $R(\theta)$ or $\theta(R)$. For surfaces with ball topology (left side of figure 1), $R$ reaches a minimum $R_0$ at $\theta = 0$ with $R'(\theta = 0) = 0$. For the surfaces with cylinder topology, the surface reaches a minimum value of $R$ at some $\theta > 0$.

### 3.1 Zero temperature

We start with the $T = 0$ case. It will be convenient to change variables as

$$R = \ell_s N^{\frac{1}{4}} e^{\frac{9x}{16}}, \tag{20}$$

so the metric becomes

$$ds^2 = \ell_s^2 N^{\frac{1}{2}} e^{\frac{9x}{8}} \left(dx^2 + d\Omega^2\right). \tag{21}$$

Then the area of an extremal surface parameterized by $\theta(x)$ in this geometry is

$$S = \ell_s^8 N^2 \omega_7 \int dx \sqrt{1 + \left(\frac{d\theta}{dx}\right)^2} e^{\frac{9x}{2}} \sin^7 \theta, \tag{22}$$

where $\omega_7 = \pi^4/3$. This extremal surface condition is then

$$\theta'' = \left(1 + (\theta')^2\right)\left(7\cot\theta - \frac{9}{2}\theta'\right). \tag{23}$$

This equation is translation-invariant in $x$, so we will have families of solutions related by translations.

We find that for large $x$, solutions oscillate around $\theta = \pi/2$ with the difference going to zero exponentially in $x$. Thus, all solutions end on an equator of the $S^8$.[3]

Decreasing $x$, we find that the magnitude of the derivative diverges at some $x_0$, indicating that the extremal surface turns around at this point and returns to infinity, unless $\theta = 0$ at this point. The latter case corresponds to our main case of interest where the surface intersects the boundary at spatial infinity on a single $S^7$ ($\theta = \pi/2$). In the other cases, the surface starts at the boundary on an $S^7$, turns around at $x_0$ and approaches the boundary again at the same $S^7$. Some examples of these surfaces are shown in figure 2.

Solutions that start at $\theta = 0$ (corresponding to initial conditions $x(\theta) = x_0, x'(\theta) = 0$ in the $x(\theta)$ parametrization) are all related to a single solution $\theta_0(x)$ for which $x_0 = 0$. We define the general such solution as $\theta(x; x_0) = \theta_0(x - x_0)$. Near $x = 0$, $\theta_0$ behaves as

$$\theta_0(x) = \sqrt{2x}\left(\frac{4}{3} - \frac{407}{1296}x + \frac{7523}{2488320}x^2 + \dots\right). \tag{24}$$

To understand the large $x$ behaviour of the solutions, we can write $\theta = \pi/2 + \epsilon_1(x)$ and work perturbatively in $\epsilon$. At order $\epsilon$, this gives

$$\epsilon'' + \frac{9}{2}\epsilon' + 7\epsilon = 0, \tag{25}$$

which has solutions

$$\epsilon(x) = Be^{-\frac{9x}{4}}\sin\left(\frac{\sqrt{31}}{4}x + \phi\right). \tag{26}$$

For the solution $\theta_0(x)$, we find $B \equiv B_0 \approx 1.12$ and $\phi \equiv \phi_0 \approx -2.70$. For the solution $\theta(x; x_0)$, we have $B = B_0 e^{9x_0/4}$ and $\phi = \phi_0 - \sqrt{31}x_0/4$. The first correction to this asymptotic solution due to terms nonlinear in $\epsilon$ go like $e^{-27x/4}$. We see that $\theta(x)$ rapidly approaches $\pi/2$ as $x$ increases.

## 3.2 Finite temperature

We can repeat the analysis for finite temperature. In this case, the area of the extremal surface is

$$S = \ell_s^8 N^2 \omega_7 \int dx\sqrt{\frac{1}{1 - \mu e^{-7x}} + \left(\frac{d\theta}{dx}\right)^2}\,e^{\frac{9x}{2}}\sin^7\theta, \tag{27}$$

where

$$\mu \equiv \left(\frac{R_H}{N^{\frac{1}{4}}\ell_s}\right)^{\frac{112}{9}}. \tag{28}$$

By a redefinition $x \to x + \ln(\mu)/7$, $\mu$ can be set to 1 in integrand, changing the prefactor as $\ell_s^8 N^2 \to R_H^8$. Thus, solutions to the extremal surface equations for general $\mu$ with minimum $x$ value $x_0$ are related to surfaces for $\mu = 1$ (where the horizon is at $x = 0$) by

$$\theta(x, x_0; \mu) = \theta(x - \ln(\mu)/7, x_0 - \ln(\mu)/7; \mu = 1). \tag{29}$$

---

[3]We could try to force the surface to have some other asymptotic behavior by placing cutoff surface with geometry $S^8$ at some radius and demanding that the extremal surface ends on a ball with a particular solid angle on this cutoff surface. However, keeping this angle fixed as the cutoff surface is taken to infinity, we would find that the extremal surface also goes to infinity in the limit. Thus, it would live completely in the high curvature region.

We will leave the parameter $\mu$ explicit for convenience; setting it to zero in the solutions will then give back solutions in the vacuum geometry. At finite $\mu$, the extremal surface equation is

$$\theta'' = \left(1 + (\theta')^2\right)\left(7\cot\theta - \frac{9}{2}\theta'\right) - \mu e^{-7x}\left(\frac{9}{2}(\theta')^3 + \frac{7}{1 - \mu e^{-7x}}\left(\cot\theta - \frac{1}{2}\theta'\right)\right). \quad (30)$$

Some solutions to the extremal surface equations are plotted in figure 3.

The asymptotic behaviour of the geometry is the same as before, and solutions again must approach $\theta = \pi/2$. To study the asymptotic behaviour of general solutions, we again define $\theta = \pi/2 + \epsilon$, and find that the equations of motion for $\epsilon$ to linear order are

$$\left(1 - \mu e^{-7x}\right)\epsilon'' + \left(\frac{9}{2} - \mu e^{-7x}\right)\epsilon' + 7\epsilon = 0. \quad (31)$$

This has solutions in terms of hypergeometric functions. To first order in $\mu$, the general solution is

$$Be^{-\frac{9x}{4}}\sin\left(\frac{\sqrt{31}}{4}x + \phi\right) + \mu B_1 e^{-\frac{37}{4}x}\sin\left(\frac{\sqrt{31}}{4}x + \phi_1\right). \quad (32)$$

Where $B_1^2 = 2B^2/227$ and $\phi_1$ is determined in terms of $\phi$. The first term reproduces the vacuum solution, while the second term gives the leading effects of the black hole on the asymptotic behavior of the surface. We see that the solution approaches the vacuum solution quite rapidly – this will ensure that the differences in extremal surface areas are localized to the region of the geometry where supergravity provides a good approximation.

**Extremal surfaces near the black hole horizon**

We have seen that the behavior of surfaces far from the black hole approaches that of the zero-temperature surfaces. To understand the qualitative behavior of extremal surfaces that approach very near the black hole horizon, make the redefinition $x \to x + \ln(\mu)/7$ so that the black hole horizon will be at $x = 0$, and then expand the extremal surface equation for small $x$. The leading terms are quadratic in $x$, giving

$$xx'' - \frac{1}{2}(x')^2 + 7\cot(\theta)xx' - \frac{63}{2}x^2 = 0, \quad (33)$$

where $x' = dx/d\theta$. Solutions corresponding to surfaces with disk topology have $x'(0) = 0$, and we find that these solutions diverge at $\theta = \pi$. The small $x$ equation ceases to be valid before this; the angle at which we need to include the terms at higher orders in $x$ becomes larger for smaller values of the $x(0)$ and approaches $\pi$ as $x(0)$ approaches $0$ (the location of the horizon). Examining solutions to the full equation for $\theta(x)$ we find that after increasing monotonically up to some angle $\theta_{max} < \pi$, the solution turns around, eventually asymptoting to $\theta = \pi/2$. Thus, the near black hole extremal surfaces hug the horizon for some time but always turn outward at some point before fully wrapping around. This is depicted in figure 3.

## 4  Regulated areas

We would like to know whether we can define a finite covariant gravitational observable via some regulated version of the area of these extremal surfaces.

## 4.1 Zero temperature

Let us consider a regulated expression for the area where we integrate out to some sphere with area $\omega_8 R_\infty^8$, or to $x = x_\infty \equiv 16/9 \ln(R_\infty/\ell_s N^{1/4})$. Then using the asymptotic form of the surface, we find that the integrand in the area functional goes as

$$dA = \ell_s^8 N^2 \omega_7 dx \left( e^{\frac{9x}{2}} - B^2 \left\{ \frac{8}{\sqrt{217}} \sin\left( \frac{\sqrt{31}}{2} x + 2\phi - \arcsin\left( \frac{\sqrt{217}}{28} \right) \right) \right\} + \mathcal{O}(e^{-\frac{9x}{2}}) \right). \quad (34)$$

Integrating to $x_\infty$, we find a regulated area

$$A(x_\infty)/(\ell_s^8 \omega_7 N^2) = \frac{2}{9} e^{\frac{9}{2} x_\infty} + B^2 \alpha \cos\left( \frac{\sqrt{31}}{2} x_\infty + \tilde{\phi} \right) + A_f(x_\infty), \quad (35)$$

where $\alpha$ is a constant and $A_f(x_\infty)$ has a finite limit as $x_\infty \to \infty$. To define a finite quantity, we can now simply subtract off the exponential term and the purely sinusoidal term.[4] Alternatively, we can define

$$A_{reg}(x_0) = \lim_{x_\infty \to \infty} \mathrm{Avg}_{[x_0, x_\infty]} \left( A(x_\infty) - \frac{2}{9} N^2 \ell_s^8 \omega_7 e^{\frac{9}{2} x_\infty} \right). \quad (36)$$

For the vacuum geometry, we can now work out this regulated area explicitly. Using $\theta(x; x_0) = \theta_0(x - x_0)$, we have

$$A_{reg}(x_0)/(\ell_s^8 \omega_7 N^2) = e^{\frac{9}{2} x_0} A_{reg}(x_0 = 0)/(\ell_s^8 \omega_7 N^2) \approx -0.98245 e^{\frac{9}{2} x_0}, \quad (37)$$

where the numerical coefficient is obtained by numerically solving for $\theta_0(x)$ and evaluating the regulated area explicitly. In terms of the minimal value $R_0$ of the coordinate defined as the proper sphere radius, we have

$$A_{reg}(R_0) = -0.98245 \omega_7 R_0^8. \quad (38)$$

We can think of the subtracted divergent piece as the area of the surface at $\theta = \pi/2$, so the negative sign here indicates that the areas are smaller than the area of the $\theta = \pi/2$ surface (i.e. the surface for $R_0 = 0$). This is natural, since we expect that surfaces further towards the boundary are associated with smaller subsystems or subalgebras of observables.

## 4.2 Finite temperature

To define regulated areas in the finite temperature geometry, we can use the same subtraction procedure as before, integrating to $R = R_\infty$, subtracting off $2/9 \omega_7 R_\infty^8$, and then averaging the resulting sinusoidal function of $x$ for large $R_\infty$. This defines a regulated area function $A_{reg}(R_0, R_H)$. By dimensional analysis, this must be $R_0^8$ times some function of the dimensionless ratio $R_0/R_H$, or equivalently, some function of the difference $x_0 - x_H$. Thus, we define

$$A_{reg}(R_0, R_H) = A_{reg}(R_0) F(x_0 - x_H) = A_{reg}(R_0) F\left[ \frac{16}{9} \ln\left( \frac{R_0}{R_H} \right) \right]. \quad (39)$$

The function $F$ compares the reguated area in the black hole geometry for some $R_0$ to the area of the surface in the vacuum geometry with the same $R_0$. We have computed this numerically for various values of $x_0 - x_H$; results for this computation are shown in figure 3. The function $F(x_0)$ approaches 1 as $x_0 \to \infty$. This is the limit in which the black hole is small, so it seems appropriate the answer approaches the vacuum result.

---

[4]Note that our procedure makes use of the spherical symmetry of the geometry and of our surfaces; it would be useful to understand better how this can be extended to more general cases.

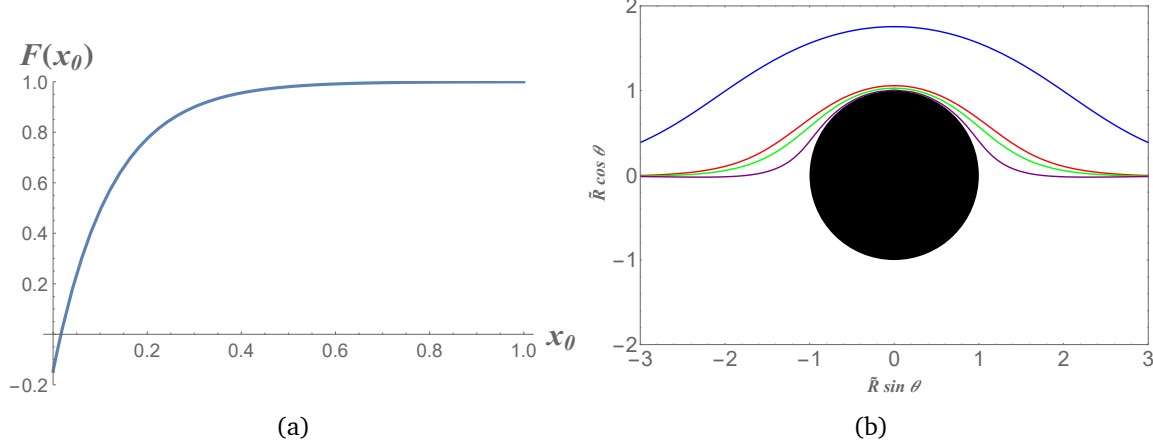

Figure 3: (a) Numerical plot of $F(x_0)$. (b) Example surfaces for $x_0 = 1, 0.1, 0.05, 0.01$ (blue, red, green, purple, respectively).

### 4.3 Validity of supergravity results

The extremal surfaces we are discussing are not confined to the region of the geometry where supergravity provides a good approximation, but asymptote into the strongly-curved region. Thus, we need to understand whether the areas we are computing are sensitive to the higher curvature corrections to the action and to the entanglement functional that are important in this region. In this section, we will see that when the black hole horizon and the minimum radial position of the surface are both well inside the region where supergravity provides a good approximation, essentially all contributions to the regulated areas we compute are also coming from this region. Thus, while the extremal surfaces enter the strongly-curved region, they all do so in almost precisely the same way, so we expect that any corrections to the extremal areas due to higher derivative terms in the equations of motion or entanglement functional will be the same for all the surfaces and can be eliminated by some modified regularization scheme.

To begin, consider the expression (35), which is also valid for the finite temperature geometries. The last term $A_f(x_\infty)$ approaches a constant for large $x_\infty$, and this limit defines the regulated area. The function approches this limit via an exponential decay. For extremal surfaces in the vacuum geometry, the leading behavior of the decay is

$$A_f(x_\infty) - A_f(\infty) \sim e^{9x_0 - \frac{9}{2}x_\infty}. \tag{40}$$

In (35), $A_f$ has been defined as a dimensionless number with the dependence on $N^2$ removed. Thus, $A_f(\infty)$ will be some order 1 number, and $A_f(x_\infty)$ will agree with this closely for $x_\infty$ large enough such that

$$e^{9x_0 - \frac{9}{2}x_\infty} \ll 1. \tag{41}$$

Recall that supergravity will be a good approximation for $R \ll N^{\frac{1}{4}}\ell_s$. Suppose we would like the regulated area functional to nearly reach its asymptotic value by some $R_\infty = \epsilon N^{\frac{1}{4}}\ell_s$ with $\epsilon \ll 1$ so that essentially all of the contributions come from the region where supergravity is valid. Suppose also that we are considering an extremal surface with some minimum radius $R_0 = \delta N^{\frac{1}{4}}\ell_s$. Then the condition (41) becomes

$$\frac{\delta^{16}}{\epsilon^8} \ll 1. \tag{42}$$

Since we have already assumed $\epsilon \ll 1$, this condition is guaranteed to be satisfied as long as $\delta < \epsilon$, i.e. if the minimum radial coordinate of the brane is well inside the region where supergravity is valid.

Next, consider surfaces in the general finite temperature geometries. Here, the leading behavior of the decay of $A_f(x_\infty)$ is

$$A_f(x_\infty) - A_f(\infty) \sim \frac{1}{5}\mu e^{-\frac{5}{2}x_\infty}. \tag{43}$$

Recalling that $A_f(\infty)$ will be some order 1 number, $A_f(x_\infty)$ will agree with this closely for $x_\infty$ large enough such that

$$\mu e^{-\frac{5}{2}x_\infty} \ll 1. \tag{44}$$

Again suppose that we would like the regulated area functional to nearly reach its asymptotic value by some $R_\infty = \epsilon N^{\frac{1}{4}}\ell_s$ with $\epsilon \ll 1$ so that essentially all of the contributions come from the region where supergravity is valid. Suppose also that the horizon radius is $R_H = \delta N^{\frac{1}{4}}\ell_s$. Then the condition (44) becomes

$$\delta^8 \left(\frac{\delta}{\epsilon}\right)^{\frac{40}{9}} \ll 1. \tag{45}$$

Assuming that $\delta < \epsilon$ (i.e. that the black hole horizon is well inside the region where supergravity is valid) this condition will always be satisfied.

# 5 General asymptotic behaviour

In this section, we analyze the asymptotic behaviour of more general solutions that do not necessarily preserve $SO(8)$ symmetry. Part of the motivation for this is to understand the data that is used to specify general extremal surfaces in the D0-brane geometries. If the regulated areas for these general surfaces correspond to some entropic quantities in the matrix model, these quantities should be labeled by the same data, so this can provide a hint in identifying the correct matrix model entropies.

For this analysis, we focus on the vacuum geometry, since the asymptotic behavior of the more general thermal geometries is the same. It will be convenient to rewrite the metric as

$$\begin{aligned}
ds^2 &= dR^2 + R^2 d\Omega^2 + \frac{175}{81}dR^2 \\
&= dx^i dx^i + \frac{175}{81}\frac{(x^i dx^i)^2}{x^2} \\
&= \left(\delta_{ij} + \frac{175}{81}\frac{x^i x^j}{x^2}\right)dx^i dx^j.
\end{aligned} \tag{46}$$

Since our expectation (according to previous references [22–24] and our analysis of the symmetrical case) is that the surface should approach some equator of the spatial $S^8$ at the boundary, we will gear our parametrization to surfaces that approach the plane $x^9 = 0$ at the boundary. We parameterize the surface using the coordinates $x^1, \ldots, x^8$ via $x^9 = T(x^1, \ldots, x^8)$. The metric on the surface becomes (up to an overall constant)

$$g_{ab} = \delta_{ab} + \partial_a T \partial_b T + \frac{175}{81}\frac{1}{T^2 + x^2}(x^a + T\partial_a T)(x^b + T\partial_b T). \tag{47}$$

To evaluate the area functional $\int \sqrt{|g|}$, we note that for an $N \times N$ matrix of the form $\delta_{ab} + A_a A_b + B_a B_b$, we can calculate the determinant by noting that the eigenvectors include

$N - 2$ vectors perpendicular to both $A$ and $B$ which have eigenvalue 1 and two more eigenvectors of the form $c_1 \vec{A} + c_2 \vec{B}$. The determinant is the product of the eigenvalues for these remaining eigenvectors. We find

$$|g| = (1 + \vec{A}^2)(1 + \vec{B}^2) - (\vec{A} \cdot \vec{B})^2. \tag{48}$$

Applying this to calculate the action, we get

$$S = \sqrt{(1 + \partial_a T \partial_a T)\left(1 + \frac{175}{81}\frac{1}{T^2 + x^2}(x_b + T\partial_b T)^2\right) - \frac{175}{81}\frac{1}{T^2 + x^2}[\partial_a T(x^a + T\partial_a T)]^2}. \tag{49}$$

It is straightforward to check that $T = 0$ is a solution to the corresponding equations of motion. We can look for solutions which approach this asymptotically by expanding the action perturbatively for large $x$. We find the quadratic action

$$S^{(2)} = \frac{8}{9}(\partial_i T)^2 - \frac{175}{288}\frac{1}{x^2}(T - x^i \partial_i T)^2. \tag{50}$$

To study solutions of the corresponding equations of motion, we switch to spherical coordinates on the $S^7$. The equations of motion become

$$\frac{9}{16}\frac{1}{r^5}\partial_r(r^7 \partial_r T) + \frac{1225}{144}T + \frac{16}{9}\Delta_{\Omega_7} T = 0, \tag{51}$$

where $\Delta_{\Omega_7}$ is the Laplacian on the sphere. The sphere Laplacian has eigenvalues $-l(l+6)$ with multiplicity

$$n_l = \frac{1}{3}(l+3)\binom{l+5}{5}. \tag{52}$$

For a mode with eigenvalue $-l(l+6)$, the radial equation is

$$\frac{1}{r^5}\partial_r(r^7 \partial_r T) + \frac{1}{9^2}(35^2 - 16^2 l(l+6))T = 0. \tag{53}$$

For $l = 0$, we reproduce our previous solution

$$T_0(r) = \frac{B}{r^3}\sin\left(\frac{4\sqrt{31}}{9}\ln(r) + \phi\right). \tag{54}$$

For $l = 1$, we have

$$T_1(r) = A_1 r + \frac{B_1}{r^7}. \tag{55}$$

The multiplicity for this eigenvalue is 8. Here, the solution with asymptotic behavior proportional to $r$ corresponds to a surface that approaches a different plane at infinity rather than $T = 0$. The multiplicity of 8 corresponds to the 8 independent directions in which we can tilt the plane. Since we are interested in the solutions that approach $T = 0$ asymptotically, we set $A_1$ to zero.

For $l > 1$, we find solutions

$$A_l r^{-3 + \frac{4}{9}\sqrt{16(l+3)^2 - 175}} + B_l r^{-3 - \frac{4}{9}\sqrt{16(l+3)^2 - 175}}. \tag{56}$$

As for $l = 1$, we need to set the coefficient $A_l$ of the growing mode to zero in order to be consistent with our ansatz. We conclude that the most general solution approaching the $T = 0$ plane for large $r$ behaves asymptotically as

$$T(r, \Omega) = \frac{B}{r^3}\sin\left(\frac{4\sqrt{31}}{9}\ln(r) + \phi\right) + \sum_{l=1}^{\infty}\sum_m c_{lm} Y_{lm}(\Omega) r^{-3 - \frac{4}{9}\sqrt{16(l+3)^2 - 175}}. \tag{57}$$

It is helpful to consider the behavior of this solution on a cutoff surface at $r = r_c$. For $A = c_{lm} = 0$, the surface intersects the cutoff sphere at an equator. For any fixed $\phi$ such that

$$\sin\left(\frac{4\sqrt{31}}{9}\ln(r) + \phi\right) \neq 0, \tag{58}$$

we can, by choosing appropriate values of $B$ and $c_{lm}$ represent an arbitrary function on $S^7$. Thus, for small values of these parameters where our analysis is valid, we have a one-to-one correspondence between our asymptotic solutions for fixed $\phi$ and small amplitude functions on $S^7$. These functions describe the location of intersection between the extremal surface and the cutoff $S^8$, so we see that we can choose an extremal surface to contain an arbitrary subset of the cutoff $S^8$ whose boundary is a small deformation of the equator.

To understand the relevance of the remaining parameter $\phi$, we recall that for solutions preserving $SO(8)$ symmetry, those with disk topology have $B = B_0 e^{9x_0/4}$ and $\phi = \phi_0 - \sqrt{31}x_0/4$ in terms of the minimum $x$ value $x_0$ ($B_0$ and $\phi_0$ are fixed order 1 numbers). Thus, the parameters describing the asymptotic solution are related by

$$\phi = \phi_0 - \frac{\sqrt{31}}{4}\ln\frac{B}{B_0}, \tag{59}$$

for solutions of disk topology (i.e. those that cap off in the interior rather than turning around and asymptoting to the equator again). Thus, in the parameter space $(B, \phi)$ the solutions with disk topology correspond to some codimension 1 subset.

It is plausible that for the more general solutions which do not preserve $SO(8)$, solutions with the topology of a disk again correspond to some codimension 1 subset in the parameter space $(B, \phi, c_{lm})$, where $\phi$ is locally determined in terms of the other parameters. To see this, consider first the space of symmetry-preserving solutions for some fixed value of $B$. This will be a one-parameter family of solutions (similar to that depicted in figure 2), with a single solution of disk topology (similar to the first solution in figure 2). We can now consider small perturbations to this solution with disk topology. Near the point of minimum radial position, the solution behaves as a plane in flat space. At length scales small compared to the curvature scales of the surface and the surrounding geometry, perturbations to the surface will be governed by the Laplace equation. The general solution that behaves smoothly at the origin (expressed as a function of the transverse coordinates $x_1, \ldots, x_8$) is

$$\Delta T = \sum_n B_{i_1 \cdots i_n} x^{i_1} \cdots x^{i_n}, \tag{60}$$

where the tensors $B$ are traceless and symmetric. This data is in one-to-one correspondence to the set of functions on $S^7$: evaluating the expression above for a unit vector $x^i$ in $\mathbb{R}^8$ gives a function on $S^7$. Conversely, any function on $S^7$ can be expressed as a linear combination of spherical harmonics, and each of these is equivalent to some homogeneous function of a unit vector that can be expressed as

$$B_{i_1 \cdots i_l}^{lm} \hat{x}^{i_1} \cdots \hat{x}^{i_l}, \tag{61}$$

for some traceless-symmetric tensor $B^{lm}$.

We see that the data describing small perturbations to disk-topology solutions near the point of minimum radius are the direction on $S^8$ corresponding to the orientation of the unperturbed solution and an arbitrary small function on $S^7$. Following one of these solutions out to the asymptotic region, we expect (assuming that the perturbation remains small) to land on one of the asymptotic solutions approaching an equator. As we have seen, these are labeled by a direction on $S^8$ defining the equator, an arbitrary small function on $S^7$, and a phase $\phi$. Since

we have one more parameter here, it is plausible that the perturbed disk-topology solutions correspond to a codimension one set of the perturbed asymptotic solutions.

We expect that the more generic solutions with asymptotic behavior given by (57) correspond to surfaces that turn around in the interior and come back out to the asymptotic region, again asymptoting to an equator that may in general be different from the original one. More generally, we might have solutions with the topology of a disk with more than one puncture, where the boundary of the disk and the boundary of each puncture asymptotes to some equator. It would be interesting to understand in general which regions of the $(B, \phi, c_{lm})$ parameter space for a given asymptotic solution correspond to which topologies.

The extremal surfaces with multiple asymptotic regions may be analogous to connected extremal surfaces in AdS homologous to some disjoint set of spatial regions. We will also briefly discuss a possible entropic interpretation for these surfaces below.

## 6 Entropic quantities in the matrix model

The extremal surface areas that we have defined and computed in the previous sections appear to be well-defined observables on the gravitational side of the correspondence. They generalize the area of the black hole horizon, which is dual in the matrix model to the entropy of the full system. In this section, we will discuss the question of whether the more general regulated extremal surface areas we have calculated correspond to some entropic quantities. A similar question has been considered for $\mathcal{N} = 4$ SYM: in [22, 24], the authors discussed possible field theory quantities that could correspond to the areas of bulk surfaces whose boundaries fill the field theory directions and end on some codimension one part of the $S^5$. In particular some of our detailed suggestions below for the field theory interpretation of areas as certain algebraic entropies are similar to suggestions in [24].

Since the BFSS model is a matrix model with no spatial extent, and its Hilbert space does not decompose naturally into a tensor product, the extremal surface areas cannot correspond to the entropy associated with a spatial subsystem, and they cannot correspond to the entropy associated with a tensor factor. On the other hand, there are more general entropic quantities that can be defined without a tensor product decomposition of the matrix model Hilbert space.

**Entropies in the ungauged model**

Before proceeding to discuss these, we note that it may be useful in this context to think about the ungauged version of the matrix model. It has been argued in [34] that the low-energy states of this model (i.e. small $E/(g^2 N)^{1/3}$ in the large $N$ limit), are the same as for the gauged theory, so that we should have the same gravity interpretation. Thus, our gravity observables could equally well correspond to observables in the ungauged model. In this case, the areas could be related to an entropy associated to some tensor product decomposition of the Hilbert space where the tensor factors are associated with individual matrix elements, or some linear combinations of these, though it is not clear that this should be the case. We will consider this possibility later.[5]

**Entropy associated with a subalgebra**

We now return to thinking about the original gauged matrix model where no natural tensor-product decomposition of the Hilbert space exists. To proceed here, we recall that it is natural

---

[5]One possibility for future study is to understand the duals of solvable $N \times N$ matrix models *without* a singlet constraint, such as [35, 36]. These models give us access to the deconfined phase at the outset.

to define an entropy associated to any sub-algebra of the algebra of observables for the model.[6]

For any quantum theory, if $\mathcal{B} \in \mathcal{A}$ is a subalgebra of the algebra of observables, given a state $\rho$ for the full system, we can define an entropy associated to $\mathcal{B}$ as follows. Choose an orthogonal basis $\{\mathcal{O}_\alpha\}$ for the algebra $\mathcal{A}$ such that $\{\mathcal{O}_\alpha | \alpha \in s_\mathcal{B}\}$ is a basis for $\mathcal{B}$. If the full density matrix can be written as

$$\rho = \sum_\alpha c_\alpha \mathcal{O}_\alpha, \tag{62}$$

then we define

$$\rho_\mathcal{B} = \sum_{\alpha \in s_\mathcal{B}} c_\alpha \mathcal{O}_\alpha. \tag{63}$$

Equivalently, we can define $\rho_\mathcal{B}$ as the unique operator in the subalgebra for which $\text{tr}(\rho_\mathcal{B} \mathcal{O}_\alpha) = \langle \mathcal{O}_\alpha \rangle$ for all operators in the subalgebra (see Theorem A.7 of [37]). In the simplest cases, the entropy associated to the subalgebra can then be computed as $S_\mathcal{B} = -\text{tr}(\rho_\mathcal{B} \log \rho_\mathcal{B})$.[7]

In appendix A, we provide an explicit example for a single qubit system, taking a general state and calculating the entropy associated with the subalgebra consisting of all operators of the form $a\mathbb{1} + b\vec{m} \cdot \vec{\sigma}$ for some fixed unit vector $\vec{m}$. Note that we have a family of subalgebras which are related by rotations. We would need something similar in the BFSS case, since each of our extremal surfaces is related to other surfaces via rotations in $SO(9)$.

**Entropy associated with a subset of observables**

We can also define an entropy associated with a more general subset of observables that does not necessarily form a subalgebra.[8] Entropies of this type have been discussed previously in the context of holography in [39–41]. Given a set of (not necessarily commuting) operators $A = \{\mathcal{O}_\alpha\}$ and a global state $\rho$, we can look for a state $\rho_A$ that maximizes the von Neumann entropy subject to the constraint that

$$\text{tr}(\rho \mathcal{O}_\alpha) = \text{tr}(\rho_A \mathcal{O}_\alpha), \qquad \mathcal{O}_\alpha \in A. \tag{64}$$

Then we can define an entropy

$$S_A = S(\rho_A). \tag{65}$$

From its definition, we have that $S(\rho_A) \geq S(\rho)$. By using Lagrange multipliers to write the condition that $\rho_A$ extremizes the entropy subject to the constraints (64), we find that $\rho_A$ must take the form

$$\rho_A = \frac{1}{Z} e^{\sum_\alpha \lambda_\alpha \mathcal{O}_\alpha}. \tag{66}$$

Note that in the special case that $A$ forms an algebra, the density operator $\rho_A$ will be in the algebra, and the definition here reduces to the one above for a subalgebra.

From the definition, it is clear that for subsets of observables $A_1 \in A_2$ we will have $S_{A_1} \geq S_{A_2}$ since in the case of a smaller set of observables, we are maximizing the same quantity subject to fewer constraints. In particular, each of these entropies is always greater than the entropy of the full state.

In appendix A, we calculate as an example the entropy for a spin 1 system associated with a subset of operators consisting of a single operator $S_z$.

---

[6]See [37] for a review.

[7]In general, for any subalgebra, there is some basis for which operators in the subalgebra are all those operators of the block diagonal form $D(M_i, n_i) = \text{bdiag}(M_1, \ldots M_1, M_2, \ldots M_2, \ldots, M_n, \ldots, M_n)$ where we have $n_i$ copies of the $k_i \times k_i$ matrix $M_i$. In this general case, the entropy is defined as $-\hat{T}r(\rho_M \ln \rho_M)$ where $\hat{T}r(D(M_i, n_i)) \equiv \sum_i \text{tr}(M_i)$. In the case where the Hilbert space decomposes into tensor factors, this more general definition is required in order to reproduce the usual entropy associated with a subsystem. In the case where all $n_i$ are 1, we get the simpler expression in the main text [37].

[8]See [38] for a recent related discussion.

## 6.1 Observables in BFSS

We have described a general set of entropic quantities that can be associated with subsets of observables. We would now like to understand which subsets of observables in the BFSS models might give rise to entropies that correspond to the regulated areas we have computed on the gravity side. There are a few useful guiding principles we can use.

First, we would like to find a natural subset of observables that is labeled in the same way as the gravitational quantities we have defined. For the areas of $SO(8)$-invariant surfaces of disk topology, the corresponding surfaces are labeled by a point on the $S^8$, corresponding to the point of minimum radial position of our surface, and by an additional parameter that can be taken to be the minimum radial position of the surface or the amplitude $B$ of the leading falloff at infinity. Alternatively, this additional parameter can be understood as being related to the size of a ball-shaped region of some cutoff surface which is contained within the extremal surface.

For more general solutions, we have argued that the surfaces of disk topology are parameterized the same data as a general function on $S^7$, which we interpreted as describing the boundary shape of a topological ball on a cutoff $S^8$ contained within the extremal surface. We would like to understand whether there are natural subsets of observables that are labeled by this same data.

A second guiding principle comes from holography. In the usual case when we compute the entropy of a spatial subsystem of some CFT, an intermediate step is the construction of a density matrix for the subsystem. This density matrix contains complete information about the physics in a certain region of the dual geometry usually called the entanglement wedge [42–44], which is bounded spatially by the bulk extremal surface. In particular, CFT operators corresponding to any bulk observable localized within the entanglement wedge have the same values when evaluated using the reduced density matrix as they have when evaluated using the full state.

In the more general definitions of entropy in terms of subalgebras or subsets of observables, the analogue of the reduced density matrix is the density matrix $\rho_A$ that lives in the algebra or whose logarithm lives in the span of the chosen subset of observables. By analogy with the subsystem case, we may similarly expect that for the subset of observables whose associated entropy gives the area of some extremal surface, the associated density matrix $\rho_A$ should contain the information about bulk physics in the entanglement wedge associated with this extremal surface. Thus, the subset of observables upon which the correct entropy is defined should include those matrix model operators which correspond to bulk observables localized in this entanglement wedge.

**Current operators in BFSS**

To proceed, it will be useful to review the connection between operators in the BFSS model and bulk fields in the corresponding supergravity solutions.

To begin, we recall the origin of the holographic duality for BFSS: the physics of a collection of D0-branes in flat-space type IIA string theory is understood to be equivalent to the physics of type IIA string theory on the background of the D0-brane solution to type-IIA supergravity. In this setup, the low-energy physics of the D0-branes corresponds to physics in the near-horizon region of the D0-brane geometry. The holographic duality arises from the fact that there is a decoupling limit where the D0-brane physics is described by the BFSS model with no residual coupling to the ambient string theory; the corresponding physics in the dual picture is the physics of type IIA string theory on the near-horizon D0-brane geometry, now decoupled from the physics in the asymptotic region.

As in the more familiar examples of holography based on conformal field theories, we expect a correspondence between light fields in the near-horizon D0-brane solutions and op-

erators in the BFSS model. To understand this correspondence, consider again the low-energy physics of a collection of D0-branes in flat space, but this time in the presence of some particular mode $\phi_\alpha^{flat}$ of the type IIA supergravity fields turned on. This will couple to a particular operator $\mathcal{O}_\alpha$ in the D0-brane theory; by tuning the strength of the supergravity mode in our decoupling limit, we can end up with the BFSS theory but now with a source for $\mathcal{O}_\alpha$. In the dual picture, the mode $\phi_\alpha^{flat}$ in the asymptotic region will extent to some mode in the full D0-brane geometry, and in particular, to some mode $\phi_\alpha^{NH}$ in the near-horizon region. In the decoupling limit that gives a source for $\mathcal{O}_\alpha$ in the BFSS model, we will have a source for $\phi_\alpha^{NH}$ in the near-horizon D0-brane geometry. In this way, we have a correspondence between certain operators $\mathcal{O}_\alpha$ in BFSS and supergravity modes $\phi_\alpha^{NH}$ in the near-horizon D0-brane geometry.

According to our discussion, the operators coupling to light supergravity fields should be those that are sourced when modes of type IIA supergravity are turned on in the presence of D-branes. These operators were derived in [45]. They are the analog of the protected operators in $\mathcal{N} = 4$ SYM theory that lie in short multiplets of the $SU(2, 2|4)$ symmetry. In that case, the operators can be constructed as superconformal descendants of single-trace chiral primary operators. Similarly, in the BFSS case, the operators descend via supersymmetry transformations from a very simple set of bosonic operators of the form

$$\mathcal{O}^{i_1 \cdots i_n} \equiv \text{STr}(X^{i_1} \cdots X^{i_n}) - \text{traces}, \tag{67}$$

where STr is a symmetrized trace, defined as the average over all permutations of the objects in the trace [46, 47]. The subtracted term involves traces of the first term over pairs of $SO(9)$ indices so that the full expression is traceless with respect to the $SO(9)$ indices, for example

$$\text{Tr}(X^i X^j) - \frac{1}{9} \delta^{ij} \text{Tr}(X^k X^k). \tag{68}$$

This ensures that the set of operators obtained by considering the various index values correspond to an irreducible representation of $SO(9)$.

In the $\mathcal{N} = 4$ theory, the precise correspondence between operators and supergravity modes can be established by matching representations of the superconformal symmetry group [48]. A similar matching between BFSS operators and modes in the near-horizon D0-brane solution, based on "generalized conformal symmetry" has been given in [49, 50].

**Entropies associated with local bulk operators**

Using the correspondence between BFSS operators and supergravity modes, we can in principle determine the BFSS operators that correspond to local bulk operators in the vacuum geometry, similar to the HKLL construction in AdS/CFT [51]. Then, for some extremal surface of disk topology in the vacuum geometry, we can associate a subset of operators which correspond to the local bulk operators between this extremal surface and the boundary. One possibility for the interpretation of the area of the extremal surface would be the entropy associated with this subset of operators, or perhaps with the subalgebra generated by this subset.

For the geometries dual to other states, there should again be some correspondence between local operators in the bulk and operators in the BFSS theory, and it may be that the area of extremal surfaces in these other geometries are again related to entropies associated with subsets of operators corresponding to local bulk operators contained within the extremal surfaces.

For extremal surfaces with the same asymptotic behavior in geometries dual to two different states, it is not clear that the construction we have just described will give the same definition of entropy. On the other hand, in standard AdS/CFT, the areas of extremal surfaces with the same asymptotics in geometries dual to different states correspond to entropies

which are defined in the same way, i.e. in terms of the same subalgebra of observables. Thus, in the BFSS context, it may be more natural to look for a definition of entropy (or a subset of observables) which only makes use of the asymptotic data describing the extremal surfaces.[9]

**Entropies defined in terms of asymptotic data**

We now consider possible subsets of BFSS operators that can be described directly in terms of the asymptotic data associated with an extremal surface. We recall that for surfaces of disk topology, this data is equivalent to the specification of a ball-topology region on an $S^8$, interpreted as the region on some cutoff surface contained in the extremal surface.

An interesting feature of the operators (67) is that the set of operators spanned by these operators is in one-to-one correspondence with the set of functions on $S^8$. To see this, recall that functions on $S^8$ can be described using coordinates $x^i$ on an $\mathbb{R}^9$ into which the sphere is embedded as the surface $\vec{x}^2 = 1$. A general function can be written uniquely as

$$f = C^{(0)} + C_i^{(1)} x^i + C_{ij}^{(2)} x^i x^j + C_{ijk}^{(3)} x^i x^j x^k + \dots, \tag{69}$$

where $\{C_{i_1 \cdots i_n}^{(n)}\}$ are a collection of traceless symmetric tensors. To any such function, we can then associate a scalar operator in BFSS as

$$f = C^{(0)} + C_i^{(1)} \mathcal{O}^i + C_{ij}^{(2)} \mathcal{O}^{ij} + C_{ijk}^{(3)} \mathcal{O}^{ijk} + \dots, \tag{70}$$

using the operators defined in (67) (up to possible $n$-dependent normalization factors).

In the algebra of functions on $S^8$, there is a natural subalgebra associated to any spatial subsystem $A$, corresponding to the smooth functions which vanish on the complement of $A$. This corresponds to a subspace $\mathcal{C}_A$ of the tensors $\{C_{i_1 \cdots i_n}^{(n)}\}$ appearing in (69). Thus, given a region $A$, we can consider matrix operators of this type built from tensors in $\mathcal{C}_A$. The set of such objects forms a subspace of the full space of matrix operators.

Now, we can think of various subsets of operators related to this subset:

1. Just this set of bosonic operators.

2. These operators and all their SUSY descendants.

3. The algebra of operators generated by one of the previous two subsets.

Alternatively, one could consider the matrix objects:

$$f(X) = C^{(0)} + C_i^{(1)} X^i + C_{ij}^{(2)} X^i X^j + C_{ijk}^{(3)} X^i X^j X^k + \dots, \tag{71}$$

without taking a trace, again built from the tensors in $\mathcal{C}_A$. The individual matrix elements are not gauge-invariant, but we can extract the gauge invariant information by taking products of traces of the whole matrix.[10]

Thus, we have various possible subsets of observables, and therefore various entropies, that can be naturally associated with a spatial region on the $S^8$. These provide candidate entropies that may be associated with the regulated areas that we have calculated in the $SO(8)$ symmetric cases.

We note that the construction here also works when the region $A$ on the $S^8$ is disconnected. In this case, the entropies may be associated with the areas of extremal surfaces described at the end of section 5 which have more complicated topologies and multiple asymptotic regions.

---

[9]An alternative closely related to the HKLL construction is to look at the algebra of operators generated by BFSS operators dual to only the local bulk operators in the asymptotic region of the entanglement wedge. This seems more likely to give the same subalgebra when applied to different states/geometries.

[10]Alternatively, we can work in the un-gauged model and consider all the operators that appear here.

## 6.2 Comparison with entanglement entropies in noncommutative field theories

In evaluating candidate entropies which might correspond to the bulk extremal surface areas, it may be useful to compare with existing proposals for entanglement entropy in quantum field theories on noncommutative spaces. We recall that a matrix quantum mechanics theory expanded about a classical background configuration of non-commuting matrices can give rise to quantum field theory on a noncommutative space. For example, when we have three matrices with background values $X_0^i = CJ^i$, where $J^i$ are generators of $SU(2)$ in the $N \times N$ irreducible representation, the fluctuations of $X^i$ and the remaining matrix fields can be identified with functions on a noncommutative sphere (a fuzzy $S^2$) [52], as we now review.

**Associating matrix fluctuations with functions on a fuzzy sphere**

We recall that the vector space $V_N$ of functions on a fuzzy $S^2$ with noncommutativity parameter $N$ is defined to be the set of functions spanned by the spherical harmonics with $l < N$. To define a map from the set of matrix fluctuations $\delta M$ to this space, we identify the action of rotation generators on $S^2$ with the action $\delta M \to [J^i, \delta M]$. Matrices $\Phi_{l,m}$ which are eigenvectors of $J^2$ and $J^z$ under this action may then be associated with spherical harmonics as

$$\Phi_{l,m} \to c_l Y_{l,m} ; \tag{72}$$

the full map between matrix fluctuations and functions on the fuzzy sphere is determined from this by linearity. Different choices for the coefficients $c_l$ correspond to different possible maps here.[11]

An equivalent way to understand this mapping is to note that a general $N \times N$ fluctuation matrix $\delta M$ can be expanded as

$$\delta M = A^{(0)} + A_i^{(1)} J^i + A_{ij}^{(2)} J^i J^j + \cdots + A_{i_1 \dots i_{N-1}}^{(N-1)} J^{i_1} \cdots J^{i_{N-1}}, \tag{73}$$

where the $A^{(k)}$ are traceless symmetric tensors with indices taking values in $(1, 2, 3)$. From this presentation, we can identify a corresponding function on the sphere,

$$f_{\delta M}(\hat{n}) = b_0 A^{(0)} + b_1 A_i^{(1)} \hat{n}^i + b_2 A_{ij}^{(2)} \hat{n}^i \hat{n}^j + \cdots + b_{N-1} A_{i_1 \dots i_{N-1}}^{(N-1)} \hat{n}^{i_1} \cdots \hat{n}^{i_{N-1}}, \tag{74}$$

where $\hat{n}$ is a unit vector and $b_k$ are a set of coefficients related to the $\{c_i\}$.[12]

The set of functions $V_N$ together with the product inherited from matrix multiplication via the map (72, defines the algebra of functions on a fuzzy $S_2$ (with non-commutativity parameter $N$), and the original matrix quantum mechanics Hamiltonian defines the Hamiltonian for a non-commutative field theory on this fuzzy sphere.

**Subsets of degrees of freedom associated with regions on the fuzzy sphere**

Various papers have considered the definition of entanglement entropy for regions of a fuzzy sphere (see, for example [54–58]). The general idea is to associate to a given region $R$ on the sphere some subset $U_N^R$ of the degrees of freedom describing the matrix fluctuations and then calculate the entropy for these degrees of freedom in the state of interest. For the full space of functions on $S^2$, a natural subspace associated with a region $R$ is the set of functions that vanishes outside $R$. We can project this to a subspace $V_N^R$ (using the natural projection that

---

[11]One natural choice [53] is to normalize the spherical harmonics and matrix spherical harmonics as $\frac{1}{N}\text{tr}\Phi_{l,m}^\dagger \Phi_{l,m} = 1$, $\frac{1}{4\pi}\int d^2x Y_{l,m}^* Y_{l,m} = 1$ and set $c_l = 1$. Another natural choice is to define unit-normalized vectors $v_{\hat{n}}$ with $\hat{n} \cdot \vec{J} v_{\hat{n}} = J v_{\hat{n}}$ (where $J = (N-1)/2$), and define the map from matrix fluctuations to functions as $\delta M \to f_{\delta M}$ where $f_{\delta M}(\hat{n}) = v_{\hat{n}}^\dagger \delta M v_{\hat{n}}$. This corresponds to the map (72) with a different choice of $c_l$.

[12]Here, the choice $b_k = (J)^k$, corresponding to the replacement $J^i \to J\hat{n}^i$ seems natural.

maps $Y_{l,m} \to 0$ for $l \geq N$) and then use the bijection (72) (for our chosen $c_k$) to define an associated subspace $U_N^R$ of matrix fluctuations. For example, it was argued in [56] using this construction that the fuzzy sphere degrees of freedom corresponding to a polar region $\theta < \theta_0$ correspond approximately to matrix elements $M_{mn}$ for the various fluctuation matrices with $n + m < N(1 + \cos \theta_0)$. Once the subset of degrees of freedom $U_N^R$ is defined, the entanglement entropy associated with a region $R$ is then defined as the entropy of the reduced density matrix for this subset of degrees of freedom.

**Comparing with the algebraic definition of entanglement entropy**

It is interesting to understand whether the entropies we have described above (or other possible algebraic entanglement entropies for the BFSS model) give rise to these previously considered fuzzy-sphere entanglement entropies when the BFSS model is expanded about a fixed classical background corresponding to some noncommutative space.[13]

As an example, consider the entanglement entropy corresponding to the region $x_3 > 0$ of our $S^8$ (whose corresponding extremal surface divides the bulk space into two symmetrical halves for a pure state). We might expect that the correct algebraically defined entanglement entropy for this case would give the entanglement entropy for one half of a fuzzy sphere when the matrix model is expanded around a background where $X^1, X^2$, and $X^3$ are set to be $SU(2)$ generators. To check this, consider the subspace of tensors $A$ chosen in such a way that the function (74) is supported only on $x_3 > 0$. We will refer to tensors in this subspace as $\hat{A}$. Corresponding to this subspace we have the subspace of classical matrices $\{\Phi_{\hat{A}}\}$ the form (73) built from restricted tensors $\hat{A}$.

First consider those matrices that are not associated with the classical background, $X^I$ for $I > 3$. According to the previous subsection, we can assign a subset of degrees of freedom associated with fluctuations of these matrices to the hemisphere $x_3 > 0$ by considering fluctuations of $X^I$ of the form (73) with the tensors $A$ in $\hat{A}$. The operators built from these degrees of freedom can be understood as those built from the basic objects $\text{tr}(\delta X^I \Phi_{\hat{A}})$. The entropy associated with the $x_3 > 0$ region of the fuzzy sphere in the noncommutative field theory may be understood as the entropy associated with the algebra generated by this set of operators, together with similar operators associated with the fluctuations of the three matrices $X^1$, $X^2$ and $X^3$.

Fluctuations of $X^1$, $X^2$ and $X^3$ are more complicated because their effective description as fields on the noncommutative emergent sphere is a gauge theory. Of these three matrix degrees of freedom, one becomes a scalar field on the sphere while the other two become spatial components of the gauge field, with the time component inherited from the matrix model. The scalar degree of freedom can be identified with $\sum_{a=1,2,3}\{J_a, \delta X_a\}$, implying that we should also have in our algebra of operators $\text{tr}\left(\sum_{a=1,2,3}\{J_a, \delta X_a\}\Phi_{\hat{A}}\right)$.

We can now compare this with our more general prescription that does not assume background values for the matrices. Here, we suggested that a natural set of operators associated to the region $x_3 > 0$ on the full $S^8$ is the set of single-trace operators of the form (70) where the choice of tensors $C^{(n)}$ corresponds to functions on $S^8$ that vanish for $x_3 > 0$. The subset of these operators that are linear in $X^I$ when the theory is expanded about a classical background involving $X^1, X^2$, and $X^3$ is the set built from tensors $C^{(n)}$ associated to functions on $S^8$ which are linear in $x_I$, independent of $x_J$ for $4 \leq J \neq I \leq 9$, and vanishing for $x_3 < 0$, where we set the matrices $X^1, X^2, X^3$ to their background values. This again gives us the operators of the form $\text{tr}(X^9 \Phi_{\hat{A}})$, with $\Phi_{\hat{A}}$ as above. Further, the set of functions on $S^8$ which are linear in $\sum_{I=4}^{9}(x_I)^2$ (and otherwise independent of $x_I$ for $4 \leq I \leq 9$) while vanishing for $x_3 < 0$ is associated with with tensors $C^{(n)}$ with the property that $C^{(n)}_{IJikj...} = \delta_{IJ} A^{(n)}_{ikj...}$ ($4 < I, J < 9$), where

---

[13]This connection was also explored recently in [26].

$A^{(n)}$ is in $\hat{A}$. Since $C^{(n)}$ is traceless, this implies that $\sum_{a=1,2,3} C^{(n)}_{aaikj\ldots}$ is proportional to $A^{(n)}_{ikj\ldots}$. This gives us, expanding around the classical background to a linear level, operators of the form $\mathrm{tr}(\sum_{a=1,2,3}\{J_i,\delta X_i\}\Phi_{\hat{A}})$.

Thus, at least at linear order around a fuzzy-sphere background, the natural set of operators built from scalar fluctuations suggested by our general prescription matches with the set of operators considered previously in computing entanglement entropies on the fuzzy sphere. We leave a more detailed comparison (including an analysis of operators describing gauge field fluctuations) to future work.

### 6.3 Testing the proposals numerically

In this section, we have presented various ideas for an entropic interpretation of the extremal surface areas in geometries dual to BFSS states. While it is unlikely that these entropies can be calculated analytically, we recall that there has already been some success in numerically calculating the entropies of BFSS states and matching to results from supergravity [15–21]. Thus, we are optimistic that the candidate entropies we have described above may also be calculated numerically and compared with the regulated areas that we have calculated in this paper. This would constitute a very detailed direct test of AdS/CFT and of the connection between entropy and geometry in quantum gravity. In particular [59] has given numerical evidence that the gauged and ungauged BFSS models agree on a low energy subspace, as argued in [34].

## Acknowledgements

We would like to thank Nikolay Bobev, Masanori Hanada, Andreas Karch and Juan Maldacena for helpful discussions and comments. This work has been supported in part by the Simons Foundation and by the Natural Sciences and Engineering Research Council. TA is also supported in part by the Delta ITP consortium, a program of the Netherlands Organisation for Scientific Research (NWO) that is funded by the Dutch Ministry of Education, Culture and Science (OCW). BW acknowledges support from ERC Advanced Grant GravBHs-692951 and MEC grant FPA2016-76005-C2-2-P.

## A  Entropy example calculations

In this appendix, we provide examples of the calculation of entropy associated with a subalgebra or a subset of operators in simple spin systems.

### A.1  Example: entropy associated with a subalgebra of operators

As an example, consider the general state of a qubit system.[14] We can write the density matrix as

$$\rho = \frac{1}{2}\mathbb{1} + \vec{n}\cdot\vec{\sigma}, \tag{75}$$

where $|\vec{n}| < 1$ and the Pauli matrices are normalized to have eigenvalues $\pm 1/2$. The entropy of this state is

$$S = -\frac{1+|\vec{n}|}{2}\log\frac{1+|\vec{n}|}{2} - \frac{1-|\vec{n}|}{2}\log\frac{1-|\vec{n}|}{2}, \tag{76}$$

---

[14]A similar example was discussed in [60].

which decreases monotonically with $|\vec{n}|$ from $\log(2)$ at $|\vec{n}| = 0$ to zero at $|\vec{n}| = 1$.

Now, we can consider the subalgebra consisting of all operators of the form $a\mathbb{1} + b\vec{m} \cdot \vec{\sigma}$ for some fixed unit vector $\vec{m}$. In this case, we have

$$\rho_{\mathcal{B}} = \frac{1}{2}\mathbb{1} + (\vec{m} \cdot \vec{n})\vec{m} \cdot \vec{\sigma} \tag{77}$$

and the entropy is

$$S_{\mathcal{B}} = -\frac{1 + |\vec{m} \cdot \vec{n}|}{2}\log\frac{1 + |\vec{m} \cdot \vec{n}|}{2} - \frac{1 - |\vec{m} \cdot \vec{n}|}{2}\log\frac{1 - |\vec{m} \cdot \vec{n}|}{2}. \tag{78}$$

This is always greater than or equal to the entropy of the full state, because of the monotonicity property noted above and the fact that $|\vec{m} \cdot \vec{n}| \leq |\vec{n}|$.

## A.2   Example: entropy associated with a subset of operators

As an example of an entropy associated with a subset of operators which is not a subalgebra, consider a spin 1 system. A general mixed state has a density matrix that can be represented in some basis as

$$\rho = \text{diag}(p_1, p_2, p_3), \tag{79}$$

for non-negative $p_i$ with $p_1 + p_2 + p_3 = 1$. The entropy here is $S = -\sum_i p_i \log p_i$. Now, consider as an example the operator $\mathcal{O}$ represented in this basis by $\text{diag}(1, 0, -1)$. The state $\rho_A$ which maximizes the entropy subject to $\text{tr}(\rho_A \mathcal{O}) = \text{tr}(\rho \mathcal{O})$ must take the form $\rho_A = Z^{-1}\exp(\alpha\mathcal{O})$. Imposing the normalization condition and our constraint, we find that

$$\rho_A = \frac{1}{1 + y + y^{-1}}\text{diag}(y, 1, y^{-1}), \qquad y = \frac{(p_1 - p_3) + \sqrt{4 - 3(p_1 - p_3)^2}}{2(1 - (p_1 - p_3))}. \tag{80}$$

The entropy associated with the subset of observables consisting of the single observable $\mathcal{O}$ is then

$$S_A = -\frac{(y - y^{-1})\log(y)}{1 + y + y^{-1}} + \log(1 + y + y^{-1}). \tag{81}$$

We can check that this is always larger than the entropy of the full state. This is also larger than the entropy associated with the subalgebra generated by $\mathcal{O}$ and the identity operator, which in this case is equal to the full entropy, since the density matrix $\rho$ lives in this subalgebra.

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
