# Peer review of "Areas and entropies in BFSS/gravity duality"

_SciPost Physics, doi:SciPost Phys. 8, 057 (2020)_

## Round 1 · Referee Report · Anonymous (Referee 1) · 2020-2-27

Strengths

1) Addresses an important outstanding question in holography by considering how entanglement of non-geometric degrees of freedom relates to geometry of gravitational theory. 2) Gives concrete proposals as to how the Ryu-Takayanagi formula might generalize to the BFSS matrix model despite lack of spatial subsystems. 3) Explicitly computes (regulated) areas of extremal surfaces preserving SO(8) symmetry in black brane background which should be dual to entropies in BFSS.

Weaknesses

1) I am confused by the emphasis placed on the gauging of the BFSS matrix model at the beginning of section 6. First off, while it is true that the U(N) singlet constraint prevents a tensor factorization of the physical Hilbert space over matrix entries, it is not clear that would have been the desired factorization relevant for spatial subregions in the bulk (i.e. the subset of operators which can then reconstruct the dual operates within the "entanglement wedge"). Secondly, the contrast with N=4 SYM seems a bit surprising, since N=4 SYM is also a gauge theory with a physical Hilbert space that doesn't spatially tensor factorize (even for a lattice regularization). Further, it is unclear the algebras proposed later in that section don't have non-trivial centers (which, in the gauge theory case, is equivalent to the lack of simple tensor factorization). 2) Section 6 could have perhaps been streamlined. While the pedagogical presentation of entropies associated to subalgebras might be useful, the one qubit examples are possibly worked out in excessive detail for how relevant they are.

Report

This paper makes concrete and original efforts to address a very important problem in holography, providing a step forward in generalizing RT to other string-based examples. While obviously very hard to compute in BFSS, it would be nice to find some testing ground for this proposal (perhaps in some simpler model), since many of the conjectures rely primarily on matching symmetries. Nonetheless, it is a precise and useful proposal relying on the parametrization of asymptotic data needed to specify these extremal surfaces.

---

## Round 2 · Author Response

Dear Editor,

We have made revisions to the paper that address the points of the
referee.

---

## Round 2 · List of Changes

For point 1), we have added a few additional comments to the first two
subsections of section 6 emphasizing that understanding entropies in the
BFSS model is challenging, both because there are no spatial subsystems
(in contrast to higher-dimensional examples of AdS/CFT) and because the
model is gauged. We also made it more clear that the obvious tensor
factors present in the ungauged model are not necessarily the right
subsystems to associate with the bulk extremal surfaces.

For point 2), we have moved the pedagogical examples of entropy
calculations for simple systems to appendix A.

You are currently on this page

Resubmission 1911.11145v2 on 7 April 2020

---

## Editorial Decision

published